# Chromatin Manipulation and Editing: Challenges, New Technologies and Their Use in Plants

**DOI:** 10.3390/ijms22020512

**Published:** 2021-01-06

**Authors:** Kateryna Fal, Denisa Tomkova, Gilles Vachon, Marie-Edith Chabouté, Alexandre Berr, Cristel C. Carles

**Affiliations:** 1Laboratoire de Physiologie Cellulaire et Végétale, Université Grenoble Alpes, CNRS, CEA, INRAE, IRIG-LPCV, 38000 Grenoble, France; kateryna.fal@cea.fr (K.F.); gilles.vachon@cea.fr (G.V.); 2Institut de Biologie Moléculaire des Plantes du CNRS, Université de Strasbourg, 12 rue du Général Zimmer, 67084 Strasbourg CEDEX, France; denisa.tomkova@etu.unistra.fr (D.T.); marie-edith.chaboute@ibmp-cnrs.unistra.fr (M.-E.C.)

**Keywords:** epigenome editing, histone marks, CRISPR-dCas9, chemical inhibitors

## Abstract

An ongoing challenge in functional epigenomics is to develop tools for precise manipulation of epigenetic marks. These tools would allow moving from correlation-based to causal-based findings, a necessary step to reach conclusions on mechanistic principles. In this review, we describe and discuss the advantages and limits of tools and technologies developed to impact epigenetic marks, and which could be employed to study their direct effect on nuclear and chromatin structure, on transcription, and their further genuine role in plant cell fate and development. On one hand, epigenome-wide approaches include drug inhibitors for chromatin modifiers or readers, nanobodies against histone marks or lines expressing modified histones or mutant chromatin effectors. On the other hand, locus-specific approaches consist in targeting precise regions on the chromatin, with engineered proteins able to modify epigenetic marks. Early systems use effectors in fusion with protein domains that recognize a specific DNA sequence (Zinc Finger or TALEs), while the more recent dCas9 approach operates through RNA-DNA interaction, thereby providing more flexibility and modularity for tool designs. Current developments of “second generation”, chimeric dCas9 systems, aiming at better targeting efficiency and modifier capacity have recently been tested in plants and provided promising results. Finally, recent proof-of-concept studies forecast even finer tools, such as inducible/switchable systems, that will allow temporal analyses of the molecular events that follow a change in a specific chromatin mark.

## 1. Introduction

Plants, as all eukaryotes, have their nuclear DNA compacted into a structured nucleoprotein complex called chromatin. Chromatin is made of a repeating subunit named nucleosome, comprising 147 base pairs of genomic DNA wrapped around a histone octamer (two molecules of each of the histones H2A, H2B, H3 and H4) [1]. Successive nucleosomes are connected via a variable length of linker DNA (i.e., from 8 to 114 bp depending on cell type and species, as well as on underlying genomic region and sequence) that typically connects to histone H1 [2]. Within an interphase nucleus, nucleosomes are not evenly distributed along chromosomes, ultimately leading to the formation of distinct functional chromatin territories. As such, we classically distinguish heterochromatin from euchromatin as it contains denser and more regularly spaced nucleosomes [3]. Besides being structurally important to enable DNA fitting into the nucleus, chromatin represents an inherent barrier for all DNA-based processes, thereby rendering all genome-related functions (e.g., transcription, replication, DNA repair and recombination) dependent on changes in histone-DNA and/or histone-histone contacts inside the chromatin, for proper access to the target DNA [4,5]. In particular, chromatin relaxation or compaction in a dynamic fashion are crucial for establishing precise, robust and/or timely gene expression patterns in order to drive developmental programs and environmental responses. These dynamic changes are achieved through different mechanisms and are categorized into different states, ranged from transcriptionally active to poised or constitutively silenced chromatin [6].

Core histones are among the most evolutionarily conserved of all eukaryotic proteins. Each core histone consists of a structured fold domain flanked by an unstructured tail (i.e., an N-terminal tail for all four histones, plus an additional C-terminal tail for H2A only). These histone tails protrude from the nucleosome core and establish specific interactions with the negatively-charged DNA, owing to their high amounts of positively charged amino acids lysine and arginine. Histone tails can be highly decorated with several different types of covalent post-translational modifications (PTMs), such as lysine (K) and arginine (R) methylation, K acetylation, serine (S) and threonine (T) phosphorylation and lysine monoubiquitination, among others. These marks are brought or removed by specific protein machineries often referred to as “writers” and “erasers”, respectively [7]. While histone acetylation directly influences chromatin structure, other marks like methylation (that can come at different degrees: mono, di or trimethylation) function as signal platforms to recruit effector modules named “readers”, which ultimately lead to a functional outcome. Combinations of histone marks define precise chromatin functional domains and their dynamics can lead to more or less direct changes in the chromatin structure and organization [8,9]. A supplementary layer of epigenetic information is brought by the different variants existing for all histones, which differ in their primary amino acid sequences and play a crucial role in establishing marks on histone tails. The most studied example is that of histone H3 variants H3.1 and H3.3 which differ only in four amino acids in *Arabidopsis* [10], and yet display specific genome-wide distribution and post-translational modifications. While H3.1 is enriched in heterochromatic regions and in silent areas of the genome containing repressive marks (H3K27me3, H3K9me3 and DNA methylation), H3.3 is enriched in euchromatic regions marked by H3K4 methylation, H2B monoubiquitination, and RNA Pol II occupancy [11,12,13].

Because histone PTMs play an important role in the regulation of gene access and expression, a complex language termed “the histone code” was coined to act in complement to the genetic code in determining the course of development and phenotypes [14]. Since then, this code undergone several readjustments and it should be seen rather as a consequence of the cumulative effect of histone PTMs than the interpretation of a real alphabet [15].

To date, the roles of PTMs on DNA activities were indirectly deduced from functional studies of their enzymatic complexes by means of mutants for their corresponding components. Unfortunately, such classical approaches have now reached their limits in defining the genuine functions of the chromatin marks themselves. Multifaceted interactions exist within and between these writer, reader and eraser complexes [16,17]. Moreover, and this property is even more represented in plants, the chromatin complexes components frequently belong to large multigene families and display functional redundancies. This, together with our imprecise knowledge of each enzyme’s specificities toward amino acids on histones (also sometimes on non-histone proteins) [18,19], has allowed drawing only limited and mainly correlative conclusions on the relationships between histone marks, transcription and chromatin function. We therefore have reached the limits of conventional molecular genetic methods to understand the precise functions of histone modifications in the context of plant growth and development. Thus, the advent of epigenome editing tools stands out as a fabulous opportunity to overcome these limits.

Here, we present the various tools and technologies developed to impact epigenetic marks, and which could be employed to study their direct impact on nuclear structure, transcriptional activity, gene expression, and their further genuine role in body plan organization in plants. We first review the epigenome-wide approaches, which include drug inhibitors for chromatin modifiers or readers, nanobodies directed against histone marks, or lines expressing histone or chromatin writers/readers mutants. A second component of this review is the report of more recent locus-specific approaches, which use proteins engineered to send a transcriptional effector or enzymatic domain to a target sequence. They rely on recognition of a DNA sequence by a protein domain [Zinc Finger (ZF) or Transcription Activator-Like Effectors (TALEs)] or by a guide RNA (CRISPR-dCas9). We present the principles of these genome-wide and target specific approaches, their proofs of concept which were mainly obtained on animal cell cultures, and further focus on their recent uses in plants. We also discuss their advantages and limitations, and how they have improved or could further implement our knowledge of chromatin marks functions.

## 2. Drug-Induced Chromatin Modifications

Chromatin modifications are essential for correct cell homeostasis and their deregulation often results in abnormal expression of key genes, causing a wide range of diseases in animals, including cancers. Because chromatin modifications are reversible, they captured researchers’ attention as potential targets for therapies. In this respect, intensive efforts have been invested in screening for natural or synthetic chemical agents able to target chromatin-related enzymes (i.e., writers and erasers), with some of them already being used in clinical trials [20]. Compared to animals, only a very small portion of known “epidrugs” have been tested in plants, which can be categorized into four groups based on the type of chromatin effector they target: (*i*) histone deacetylase inhibitors, (*ii*) histone acetyltransferase inhibitors, (*iii*) histone methyltransferase inhibitors and (*iv*) molecules that disrupt methyl supplies for methylation reactions (Table 1). A key drawback for epidrugs is their potential lack of specificity toward a single enzymatic activity. Indeed, some of these compounds (such as nicotinamide, HC toxin, nitric oxide and curcumin) have natural origins and most likely pleiotropic functions. However, recent progress in the knowledge of protein structures and chemical synthesis allowed designing active compounds to target active centers of the selected enzymes, giving rise to a range of synthetic inhibitors such as RDS 3434 and BIX-01294. Another drawback for epidrugs is the need for a treatment to be supplied within the growing medium, by spray, or by injection, which does not permit to easily target specific tissues or cell types, particularly in plants. Therefore, technologies have been developed that allow perturbators of chromatin effectors to be expressed by the cell or/and within the organism. A first efficient category of tools are the nanobodies.

## 3. Nanobodies for Exploration and Modification of Histone Marks

Nanobodies are synthetic derivatives of the variable antigen-binding region (VHH) from heavy-chain-only antibodies naturally produced in camelids and sharks [21]. Their small size (~15 kDa), high stability and strong binding affinity make nanobodies promising tools for both basic and clinical research [22]. For instance, a number of studies have been dedicated to development of nanobody-based drug delivery systems for treatment of a wide spectrum of human diseases [23]. Besides medical applications, fluorescent modification-specific intracellular antibodies named mintbodies, were designed to track in vivo H3K9 acetylation during Drosophila embryogenesis and in zebrafish [29]. Mintbodies provide the unprecedented advantage to follow histone mark dynamics in live systems thanks to endogenously expressed antibodies, as opposed to classical immuno-detection approaches that require the use of exogenous antibodies on fixed tissues or cells. However, very few instances of nanobodies recognizing specific histone marks have been reported thus far. As a matter of fact, an alpaca-derived nanobody designed to recognize the γ-H2AX (i.e., a reliable biomarker of DNA double-strand breaks) was reported as a potent tool to detect DNA damages in vitro and eventually in vivo, assuming that alternative epitope recognition and epitope masking may limit its application [30]. Similarly, a nanobody initially directed against phosphorylated γ-H2AX but actually targeting H2A-H2B heterodimers (referred as “chromatibody” below) was successfully used for in vivo high-resolution dynamic chromatin/chromosome imaging in human cells and in Drosophila [31]. Genetically encoded chromatibodies have the great advantage over tagged histones that they allow detection of the histones encoded by endogenous genes and not expressed from a transgene. In addition to tracking chromatin dynamics in living cells, nanobodies may be used to target specific enzymatic activities at the nucleosomes, thereby inducing genome-wide changes in a given epigenetic mark. Such a strategy was proven efficient in human cell cultures where a chromatibody-fused RNF8 (Ring Finger Protein 8) E3 ubiquitin ligase induced an increase in histone monoubiquitination at the whole-genome scale [31]. While nanobodies are no doubt a potent powerful tool for chromatin-related studies, there is currently no such report in plants. Indeed, nanobody-based technologies were thus far used for protein tracking, re-/mis-localization, purification/crystallization, degradation and modulation in the model plants *Arabidopsis* and tobacco, as well as to detect plant toxins and pathogens or to mediate resistance against plant pathogens [36].

## 4. Direct Gene Manipulation for Genome-Wide Chromatin Modulation

The consequences of genome-wide chromatin modulation have initially been explored using classical genetic strategies. One consists in studying mutants, in particular T-DNA insertion lines, for chromatin factors or for histone encoding genes. Another one consists in expressing histone genes mutated at precise residues, with amino acid substitutions that prevent a mark of interest to be deposited. Finally, the more recent advent of the Clustered Regularly Interspaced Palindromic Repeats (CRISPR)-CRISPR-Associated (Cas) technology allows precise editing of chromatin modifier enzymatic modules or of histone residues. Such approaches remain however limited as they do not directly edit epigenetic marks, and thus rather allow only correlative conclusions on the function of histone modifications. We nevertheless briefly describe the timeline of their use, starting with classical genetic approaches and then transitioning to the CRISPR-Cas9-based editing technology.

### 4.1. The Limitations of Knock-Out and Knock-Down Mutant Lines for Mark Function Analyses

One classical genetic strategy to address the function of chromatin mark dynamics focusses on histone-modifying enzymes using corresponding classical loss- and/or gain-of-function mutants. Salient examples in plants are those of mutants in the components of the Polycomb group (PcG) Repressive Complexes 1 and 2 (PRC1 and PRC2), or mutants in the trithorax group (trxG) family members, or in enzymes involved in H2B mono-ubiquitination [37]. However, because these enzymes can have multiple and/or non-histone substrates and because multiple enzymes exist that bring the same modification to the same residue target, results from such a strategy are typically only suggestive of true histone marks functions.

Another classical genetic strategy studies mutants in histone encoding genes. Two reports in *Arabidopsis* show the effect of multiple mutations in *HISTONE THREE RELATED* (*HTR*) genes combined with inducible artificial micro RNAs (amiRs) to knock-down the expression of H3.1 and H3.3 histone variants [39,40]. They reveal the role of H3K27me3 in the regulation of flowering time, maintenance of inheritable silencing during cell division, as well as its effect on H3 variants dynamics. Moreover, the conditional expression of the H3 variant H3.10, normally specific to sperm cells, revealed that it is immune to lysine 27 methylation [43]. Its use in other cellular, tissue, or developmental contexts would allow to study the function of methylation at H3K27. Work on H2A variants also helped in elucidating specific histone variant function on gene expression. While loss-of-function mutations for the H2A.Z deposition complex (a H2A variant associated with highly expressed genes) lead to reduced expression of flowering related genes such as *FLOWERING LOCUS C* (*FLC*), *MADS AFFECTING FLOWERING 4* (*MAF4*) or *MAF5* [44], a more direct approach using a triple *h2a.z* mutant line showed no change in DNA methylation level but strong mis-regulation of developmental genes leading to defects in flowering time and floral homeotic transformations [46]. This latest strategy, while it directly relates to histones, still does not provide precise insights into the function of a specific chromatin mark. Other approaches more focused on specific modifiable residues, consist in expressing histones mutated at specific amino acids to prevent their post-translational modification. We review examples of successful studies in Section 4.2.

Moreover, the use of T-DNA insertion lines can sometimes be misleading, in particular due to chromosome rearrangements, which may have impacts on the epigenome [47]. This was the case with the study of *h2a.w* double and triple knock-out mutants of *Arabidopsis*, which indicated that the H2A.W variant that specifically localizes in heterochromatin, is required for its condensation and is essential for plant growth and fertility [49]. Actually, a large genomic rearrangement in one of the T-DNA insertion mutant allele hindered proper functional analysis of H2A.W: the initially reported severe developmental and heterochromatin defects, rather than being caused by the loss of H2A.W function, were due to a duplication of the *CMT3* locus. This was revealed by a newly constructed CRISPR-Cas9-induced null *h2a.w* triple mutant, which displayed no visible developmental phenotypes and had only minor effects on gene expression [51]. The CRISPR-Cas9 editing tool thus brings new perspectives, notably avoiding T-DNA-induced wide genomic rearrangements with unexpected effects on the epigenome. Attempts in generating CRISPR-Cas9-induced mutants in histone or chromatin factors are reported in Section 4.3.

### 4.2. Expressing Mutant Histones Carrying Non-Modifiable Residues

A straightforward way to study the impact of chromatin marks is to express histones in which a certain modifiable amino-acid residue is substituted by a non-modifiable one. Such mutants can be expressed in an ectopic, conditional or inducible manner. This approach was extensively applied in animals, yeast and fungi over the last decade [53,56,57] to explore for example crosstalk between histone modifications or because these mutations, often named “oncohistone” mutations, have been linked to cancers [59]. K-to-M mutations in histone H3, associated with developmental and cancer pathologies in humans, have already been studied in detail in the animal field, revealing the role of methylation at histone lysine residues in promoting HMT enzymatic activity [24,61,63,65]. In particular, the functions and mechanism of spread for the H3K27me3 mark were deduced from structural comparative analyses involving H3K27 and H3M27 peptides [25,26]. A study on transgenic mice with inducible K-to-M mutations revealed the specific roles of methylation at H3K9 and H3K36 in chromatin accessibility, gene expression landscapes and their reversible effects in differentiation programs [53]. Another recent K-to-M substitution experiment revealed that lysine 4 of histone H3.3 is required for embryonic stem cell differentiation, histone enrichment at regulatory regions and transcription accuracy [27].

In plants, the implementation of this strategy is much more recent. A transgene carrying a H3.3 K-to-M substitution at lysine 36 and expressed in *Arabidopsis* wild-type plants acts in a dominant-negative manner to cause a global reduction of H3K36 methylation and strong developmental perturbations resembling those observed with the H3K36 methyltransferase SDG8 loss-of-function mutants [28,32].

Even if the approach using mutations in histones seems rather direct, a limitation resides in the existence of non-allelic histone variants (e.g., the canonical H3.1, the variant H3.3, the centromeric CENP-A/CENH3…) often encoded by multigene families (e.g., the *Arabidopsis* genome contains five H3.1 encoding genes), which renders the direct manipulation of histone primary amino acid sequence difficult and incomplete. Moreover, even though informative on the role of a specific amino acid, this approach presents some limits in driving precise conclusions on the effect of a specific mark, because a given histone residue may carry different types of modifications. As examples, lysine 9 and 27 of histone H3 can be methylated which correlates with gene repression, or acetylated which correlates with active expression.

### 4.3. Editing Histones and Chromatin Factors: New Perspectives Brought by the CRISPR-Cas Tool

First discovered in *Escherichia coli* as an adaptive immune system, the CRISPR-Cas system is now the most widely used system for genome-editing applications. This system works on a recognition-cleavage-memory acquisition basis, where a bacterial CRISPR guide RNA (gRNA) recognizes a foreign invading DNA and a CRISPR-associated endonuclease (Cas) cleaves the foreign DNA to neutralize the invader. The cleaved DNA is incorporated into the CRISPR locus, providing a “genetic memory” against later potential infections. There are two main classes of CRISPR-Cas systems, with the most commonly used in genome engineering being a Class 2 ribonucleoprotein complex from *Streptococcus pyogenes*, consisting of a Cas9 DNA endonuclease associated with a two-part single-gRNA (sgRNA). As its name suggests, the sgRNA is a single RNA molecule containing the custom-designed short CRISPR RNA (crRNA) fused to the scaffold trans-activating crRNA (tracrRNA) providing the stem loop structure for CRISPR nuclease binding. The double-stranded DNA target can be recognized by the sgRNA if a near-perfect base-pair complementarity exists between the target DNA strand and a 5′-terminal 20 nucleotides sequence, or “seed” region, in the crRNA. In addition, the recognition will only occur if the DNA target contains a protospacer adjacent motif (PAM; 5′NGG-3‘, where N is any nucleotide) immediately downstream of the target site. Then, upon sgRNA-target DNA hybridization, the endonuclease Cas9 generates site-specific breaks in the double-stranded DNA target [33].

In recent years, programmed sgRNA have been repurposed to efficiently send the Cas9 endonuclease at specific genomic loci in eukaryotic cells. Cas9 produces double-strand breaks which can be exploited to introduce genetic modifications through the recruitment of DNA repair mechanisms. Compared to ZF or TALE-based editing tools, the only variable is the small RNA guide that needs to be specifically designed by identifying PAM sites in the genome adjacent to the highly specific 20 bp target sequence [34] (Figure 1A,B). For that, numerous in silico tools exist such as CHOPCHOP [35], CRISPRdirect [38], GT-Scan [41] or CRISPRseek [42], Cas-OFFinder [45] or Off-Spotter [48]. Afterwards, the chosen gRNA is cloned in a suitable expression vector.

As highly specific genomic scissors, the CRISPR-Cas9 system was used as an efficient editing tool to mutagenize a considerable number of genes. Yet, very few studies reported editing of histones, very likely due to the fact that some of them are encoded by multigenic families [10], thus rendering challenging the targeting of all genes coding for a given histone variant. Nevertheless, already reported highly efficient genome editing, with examples of sextuple mutants in *Arabidopsis* [50], is promising in that sense. Another limitation for applying the Cas9 editing approach on histone coding genes is the potential lethality of histone mutants, as shown for H3.1 and H3.3 variants [39,40]. A report describes the editing of histone H2A variants (H2AX and macroH2A, involved in DNA Damage Response) using CRISPR-Cas9 in human cells [52]. In plants, the above mentioned CRISPR-Cas9-induced new null *h2a.w* triple mutant (part 4.1) showed that H2A.W fine-tunes the accessibility of heterochromatin by preventing deposition of the linker Histone H1, thereby facilitating access to non-CG DNA methylation factors [51].

All other reports on chromatin components editing concern genes encoding proteins involved in writing, erasing or reading epigenetic histone marks (the so-called “chromatin effectors”). In animals, CRISPR-Cas9 was successfully used to generate mutations in the key PRC2 component EZH2 (enhancer of zeste homologue 2) responsible for the deposition of the repressive mark H3K27me3 and involved in several diseases [54,55]. Similarly, the H3K9me3 reader and chromatin remodeler ATRX was genetically inactivated by CRISPR-Cas9 to explore its role in regulating glioma malignancy and chemoresistance [58]. In plants, several studies of edited chromatin effectors were published. In poplar, the H3K9 demethylase JMJ25 was knocked-out by CRISPR-Cas9 to study anthocyanin synthesis and pathway [60]. In rapeseed, 2 homologs of the *Arabidopsis* major H3K36 histone methyltransferase SDG8 were characterized by creating knockout or knockdown mutants using the CRISPR-Cas9-mediated genome-editing system [62]. Interestingly, Cas9 editing efficiency is optimal at 37 °C, which corresponds primarily to the body temperature of the animal host, and strongly hampered by hypothermia treatment [64]. Because plants are grown and transformed at ambient temperatures (e.g., 20 to 25 °C), a heat shock treatment at 37 °C is therefore classically applied for improving Cas9 editing [66,67,68]. Also, it is important to limit the period of high temperature to prevent impaired growth and reproduction. Cas12, another member of the class 2 systems, recently gained attention as an editing tool, notably in plants. Indeed, it reaches optimal activities at around 28–29 °C, which is more compatible with plants growing conditions [67]. In the future, it will be highly valuable to engineer Cas variants or to identify Cas orthologs that are more active at lower temperatures. In this sense and even if it is not clear yet whether it is functional, the finding of a type II CRISPR-Cas system in the genome of *Pedobacter cryoconitis*, a facultative psychrophile from Antarctica, represents a first step [69].

## 5. Programmable DNA-Binding Platforms to Modify Chromatin at Specific Loci

A new trend in manipulating chromatin and epigenetic pathways is based on synthetical systems involving proteins that target DNA with a programmable sequence specificity. These proteins, when fused to effector proteins or catalytic domains, allow to manipulate chromatin, at specific targeted loci. So far, three types of DNA-binding platforms have been developed to engineer chromatin: (*i*) ZF proteins, (*ii*) TALEs and (*iii*) dCas9, corresponding to a modified version of the CRISPR-Cas system, in which the endonuclease domain of Cas9 is inactive due to point mutations [70]. The two first approaches are based on the property of a protein domain to recognize a specific DNA sequence, thus requiring to design a DNA binding domain sufficiently specific to target the desired genomic site (Figure 1A,B). By contrast, dCas9, because it operates through RNA-DNA interaction, only requires to design a sgRNA for recruitment of an effector to a specific genomic region (Figure 1C). These systems have been extensively used in a wide variety of applications in animals (e.g., gene regulation, genome and epigenome engineering, diagnostic applications and therapeutic options). In plants, their use is much more recent and ZFs and TALEs have been mainly employed to recruit classical transcriptional factors for direct manipulation of gene expression, while enzymatic domains that modify chromatin have rather been used in combination with dCas9. For this reason, the dCas9 strategy will be treated separately in a dedicated paragraph.

### 5.1. Zinc Finger (ZF) Engineered Proteins

The first generation of widely used DNA-binding domain (DBD) in epigenetic engineering is the ZF domain. ZF proteins are among the most common and studied groups of DNA-binding transcription factors in eukaryotes. The ZF is a small folded motif that binds one or more zinc ions to stabilize its structure through cysteine (C) and/or histidine (H) residues at the recognized DNA site [71,72]. ZF modules can be linked together in a polydactyl system of 3, 4 or 6 fingers to target DNA sequences that contain a series of DNA triplets [73]. Theoretically, a polydactyl protein containing six ZF domains should be enough to recognize a unique 18 bp target DNA sequence in the 3,3 Mbp human genome [74]. In plants, the same should be true for tobacco (~4.5 Gb), wheat (~17 Gb) or rape (~1.13 Mb), while for *Arabidopsis* (~135 Mbp), tomato (~950 Mbp) or rice (~430 Mbp) a polydactyl system of five ZF domains would be sufficient.

ZF Nuclease is the oldest genome editing technology taking advantage of the DNA sequence specificity of ZF DBD together with the non-specific cleavage activity of FokI endonuclease [75]. In order to manipulate gene expression in animal cell lines, ZFs were later used to create powerful Artificial Transcription factors (ATFs), through fusions with strong activation domains derived from Herpes Simplex Virus (VP16 and VP64) [76] or with the mammalian KRAB repressor domain [77]. Both types of domains recruit chromatin modifying enzymes, thereby inducing changes in epigenetic marks [78]. For example, VP64 recruits the histone acetyltransferase p300 which causes increase in activating H3K27 acetylation at the targeted locus [79], while repression via KRAB induces long-range spreading of repressive chromatin marks such as H3K9me3 [80]. In parallel, ZF were also used to block the binding of endogenous transcription factors at specific sites [81,82]. In plants, ATF strategies, using activators such as VP16 or VP64 [76,77]) as well as repressors such as KOX [76] and ERF-associated amphiphilic repression (EAR) [79] domains, were successfully used to modify the expression of targeted genes.

ZFs can also be fused with specific epigenetic modifiers to modulate chromatin marks at specific genes. As a potential cancer therapeutic approach, ZFs were engineered to re-activate hypermethylated DNA targets through fusion with the human Ten-Eleven Translocation (TET1) DNA demethylation inducer [83,84]. ZFs were also used to epigenetically repress target genes in human cell lines using the catalytic domains of DNA or H3K9 methyltransferases [85,86,87,88]. However, the achieved effect was only transient due to the loss of methyltransferases expression, which occurred within days, as a result of the chosen delivery method based on adenoviral vectors. For this reason and also because epigenetic marks are part of complex and multivalent epigenetic networks, authors discussed that an epigenetic mark cannot be reset easily. Despite these limitations, the ZF-induced epigenome editing approach was also validated in plants. Firstly, tethering the SET- and RING-ASSOCIATED domain-containing SUVH2 protein with an artificial ZF to an unmethylated site was shown to be sufficient for recruiting RNA Pol V and establish DNA methylation and further gene silencing [89]. Moreover, a fusion of the TET1 catalytic domain with an artificial ZF was designed to target either the promoter of the *FLOWERING WAGENINGEN* (*FWA*) gene or the CACTA1 transposon. In both cases, ZF-TET1 fusions caused targeted removal of 5-methylcytosine (5mC) with high specificity and minimal off-target effects resulting in gene expression activation [90]. Secondly, an artificial ZF was used to identify diverse RNA-directed DNA methylation (RdDM) components able to promote DNA methylation and silencing at an unmethylated epiallele of *FWA*, as well as at thousands of additional loci [91]. The output of this study was promising for further exploration in other systems such as CRISPR-dCas9, which are more adapted to multiplex targeting.

Nowadays, the design of specific zinc finger proteins, assembly of ZFs in a polydactyl system and the search for target sites is made easier through online tools such as the “Zinc Finger Tools” [92] or the Zif-BASE database of ZF proteins [93], and the assembly of ZFs in a polydactyl system can be done using the SuperZiF system [73]. Moreover, cell-based methods exist to efficiently and rapidly interrogate and select ZF-DNA interactions [94,95]. Engineered ZF proteins nevertheless have limits. Indeed, neighboring ZFs can interact with each other, thus affecting orientation and specificity of binding, and making prediction difficult and off-targeting more frequent [75,96]. In addition, DNA binding efficiency and activity of ZF-based systems are not necessarily correlated. Indeed, while thousands of off-target events were obtained due to sequence similarity to the on-target site, only ∼10% of bound promoters showed expression changes without clear correlation with histone marks changes [97].

### 5.2. Transcription Activator-Like Effectors (TALE)-Based Editing Tools

An alternative epigenome editing tool also using customizable DBD is the TALE system, derived from a family of effectors secreted by the bacterial plant pathogens of the genus *Xanthomonas*. Each TALE protein contains a single DNA binding domain whose specificity to a particular base pair in the target DNA sequence relies on a repeat-variable di-residue [98,99].

Like for ZFs, TALE DBD were fused with different transcriptional activators [100,101] or repressors [102] to form ATFs used in mammalian cell lines. In *Arabidopsis*, TALE DBD was successfully used to generate a chimeric transcriptional repressor that targets an element in the promoter of the drought-induced gene *RD29A* [103]. In addition, a multiplex TALE activation (mTALE-Act) system based on the VP64 activator was developed to facilitate the simultaneous activation of multiple genes [104]. More recently, a TALE-based two-component AND-gate system named split-TALE (sTALE) was shown to successfully induce the genes involved in production of the diterpene Z-abienol in *Nicotiana tabacum* [105]. In addition, TALE DBD have also been used for epigenetic editing, but thus far exclusively in animal cell lines. A collection of epigenetic mark-modifying TALE-histone effector fusion constructs (epiTALEs) were evaluated in a mouse neuroblastoma cell line for their ability to repress the transcription of two neuron-specific genes [106]. Among the 32 repressive histone effector domains tested, the epiTALE fusion with the *Arabidopsis* H3K9 methyltransferase KRYPTONITE (KYP/SUVH4) was able to promote H3K9 monomethylation and transcriptional repression of the target loci. With the intention to identify enhancer target genes, TALEs fused to the H3K4 lysine-specific demethylase 1 (LSD1) were designed to recognize nucleosome-free regions of 40 candidate enhancers in a human cancer cell line [107]. Out of 9 selected TALE-LSD1 fusions, 4 caused downregulation of a nearby gene, correlating with reduced levels of H3K4 methylation and H3K27 acetylation, thus indicating a generalized chromatin inactivation.

Compared to ZF, the TALE system is preferably used for high-throughput studies, as its design is simpler and the level of off-targets lower [98,99,108,109,110]. As for ZF, numerous online tools are available for TAL effector design and target prediction (e.g., the TAL Effector-Nucleotide Targeter (TALE-NT) 2.0 [109,111] or the Mojo Hand [112]). Nevertheless, TALE presents few important limitations, among which the variation of activity levels on target sites often observed between different TALEs designed using the same cipher [113]. One possible explanation could be related to epigenetic repression and/or inaccessible chromatin at certain loci in the endogenous genome, as supported by the valproic acid and 5-aza-2′-deoxycytidine treatments tests on the embryonic stem cells [100]. Reciprocally, DNA methylation impeded TALEs access to DNA [114]. Finally, like for ZF, the extremely rapid development and wide usage of CRISPR-Cas-derived technologies have significantly slowed down the use of TALE, especially in plants.

## 6. Targeted Epigenetic Editing with dCas9

The CRISPR-Cas9 system has revolutionized genetic engineering. Beyond this, the CRISPR-Cas9 system has also been modified to specifically modulate gene expression without altering the DNA sequence itself. To do this, a catalytically inactive or “dead” form of Cas9 (dCas9), unable to cleave target DNA, was created by introducing point mutations (D10A and H840A) into the endonuclease domains [70]. While lacking the endonuclease activity, dCas9 can still bind a gRNA and target a DNA sequence in the genome with the same precision. The dCas9 system rapidly emerged as a programmable DNA-binding platform that distinguishes from the ZF and TALE approaches by its specificity, adaptability and also reversibility. The dCas9 systems can be sorted into two generations, a first one based on simple fusions between dCas9 and effectors (historically starting with transcription factor domains), and a second one involving chimeric dCas9 systems for enhanced targeting and effector capacity (Figure 1C–E). After detailing the principles, advantages and drawbacks of the various types of systems of the two generations, we will present successful examples of epigenetic editing in both animals and plants.

### 6.1. The First Generation of dCas9 Targeting Tested with ATFs

Early experiments demonstrated that a dCas9-gRNA complex alone targeted to the coding DNA strand of a protein-coding region is sufficient to sterically block transcription elongation and/or initiation [70,115]. However, this CRISPR interference (CRISPRi) approach achieved (if at all) only modest repression in mammalian cells and could trigger the production of new antisense transcripts [70,116]. To affect transcription more efficiently, transcriptional repression or activation domains were fused to the C-terminus of dCas9. When targeted to promoter regions by a given specific gRNA, repressor domains such as KRAB and SID (SIN3-interacting domain) or activator domains such as VP64 or p65 efficiently modulated the transcription of downstream target genes in animal and yeast [117].

Single gRNA-targeted transcriptional activation or repression was also achieved in tobacco, *Arabidopsis*, maize, as well as in rice protoplasts, by fusing the dCas9 to VP64, the plant-specific transcriptional activation EDLL motif (from APETALA2/ETHYLENE RESPONSIVE FACTOR), the TAL domain or the EAR motif-containing repressor domain SRDX [118,119,120,121].

Another advantage of using dCas9-based ATFs over ZF or TALE strategies relies on its scalable multiplexing capabilities. Two mutually non-exclusive multiplexing strategies using the conventional gRNA architecture have been developed to improve dCas9-associated ATFs. The first strategy was achieved by fusing six TAL domains and two VP64 in tandem with dCas9. The obtained dCas9-derived transcriptional activator named dCas9-TV conferred far stronger transcriptional activation than the routinely used dCas9-VP64 activator, in *Arabidopsis*, rice and human cell lines [122]. The second strategy is based on the synergistic action of multiple gRNAs that are expressed simultaneously to all target the same locus [104,119,120]. In addition, dCas9 can be co-expressed with multiple gRNAs to simultaneously bind and regulate different target genes, as shown in *Arabidopsis* protoplasts co-expressing dCas9-TV with three gRNAs targeting *WRKY30*, *RECEPTOR LIKE PROTEIN* 23 (*RLP23*) and *CONSTITUTIVE DIFFERENTIAL GROWTH* 1 (*CDG1*) [122]. While these multiplexing strategies first represented promising improvements, they also revealed some drawbacks. For example, the use of multiple gRNA increases risks for off-targeting, and the dCas9-TV system showed expression and/or stability issues when more than two VP64 moieties were used [123]. To circumvent these problems and further enhance the effect of the dCas9-ATF manipulation, a second-generation of dCas9-based gene regulation with more complex architectures was developed.

### 6.2. Chimeric dCas9 Systems with Enhanced Targeting and Transcriptional Effector Capacity

The analysis of the crystal structure of the Cas9-sgRNA-DNA complex revealed that the tetraloop and the two or three stem loops of the gRNA protrude outside of the Cas9-gRNA complex, thus making them potentially usable for the recruitment of effector domains [124]. Based on this assumption, a first system was developed in which orthogonally acting protein-binding RNA sequences derived from bacteriophages, usually MS2 (Figure 1D) or PP7, were added to these loops. The resulting scaffold RNA (scRNA) can then be recognized by specific RNA-binding phage coat proteins, MS2 Coat Protein (MCP) or PP7 Coat Protein (PCP) respectively, fused to a transcriptional effector (e.g., VP64 for activation or KRAB for repression) [125]. This technology was successfully applied in mammalian cells and yeast, and was found to be more efficient than the direct dCas9-VP64 fusion system [126,127]. However, the induction of the targeted gene was found decreased when more than one copy of an hairpin was used in the scRNA [127]. Based on the same principle, the Casilio system (based on the eukaryotic Pumilio protein property to bind a specific 8-mer RNA sequence) combines three elements: (*i*) a dCas9, (*ii*) a scRNA made of a gRNA linked at its 3′ end to one or more Pumilio/FBF (PUF)-binding site(s) and (*iii*) PUF domains fused to one or more effector(s) [128]. The main advantage of the Casilio system over the “Scaffold” one is that multimerization is not a limiting factor since the linear structure of PUF-binding sites does not impede sgRNA transcription and/or dCas9/sgRNA DNA-binding capacity [128]. Also, it is worth noting that whereas a Cas9 protein fusion only allows one type of transcriptional regulation (i.e., activation or repression), the scRNA, by its modularity, could bring various activities to target loci (e.g., thereby mediating simultaneous activation and repression at target genes) [127,128]. The chimeric Synergistic Activation Mediator (SAM) system combines a single direct fusion of a transcription regulation domain to dCas9 (i.e., dCas9-VP64) with MS2 scRNA aptamers to recruit MCP fused activation domains (HS1 and p65). It allowed to induce the highest gene activation level, even with a single gRNA [126,129], however, it appeared not more efficient than other systems when activating multiple genes at once [130]. In order to recruit more effectors, another system has been developed which contains multiple copies of the short epitope GCN4 fused to dCas9. Each of the effector domains is then fused to a cognate single-chain variable fragment antibody directed against GCN4 (Figure 1E) [131]. Originally developed for imaging of single protein molecules in living cells and named SUperNova (SunTag) with reference to the very bright stellar explosion, this system was also used as a versatile platform for multimerizing proteins to create potent synthetic transcription factors [132]. While allowing efficient changes in gene expression in various cellular contexts including human, mouse, and Drosophila models, the performance of each of the above mentioned systems varies greatly depending on target loci and cell types [133]. Many more variants have been designed since, including the three-component repurposed technology for enhanced expression (TREE) system, which combines SunTag with the RNA tethering system used by SAM in a tree-resembling structure [134].

In *Arabidopsis*, a SAM system combining a dCas9 fused to VP64 and several MS2 scRNAs that recruit VP64- or EDLL-MCP, successfully achieved varying degrees of transcriptional activation of seven different genes, including the low expressed *PRODUCTION OF ANTHOCYANIN PIGMENT 1* (*PAP1*), the imprinted gene *FERTILIZATION-INDEPENDENT SEED 2* (*FIS2*) and the silenced microRNA miR319 [104]. In *Arabidopsis* transgenic plants, a redesigned SAM system using a scRNA to recruit the p65 transactivating subunit of NF-kappa B and a heat-shock factor 1 (HSF) activation domain was also able to moderately increase the expression of two endogenous genes chosen for the obvious phenotype resulting from their overexpression, *PAP1* and *Arabidopsis thaliana vacuolar H^+^-pyrophosphatase* (*AVP1*) [135]. More recently, different SunTag VP64 constructs were reported as being able to mediate the highly specific activation of several loci such as the DNA methylated and silent *FWA* gene, the two unmethylated and lowly expressed genes *CLAVATA3* (*CLV3*) and *APETALA3* (*AP3*), as well as two different *ATCOPIA93* retrotransposons, one in euchromatin (*Evadé*) and one in heterochromatin (*Attrapé*) [136]. More interestingly, the targeted activation of *FWA* was able to reduce promoter DNA methylation. Facing the exponential development of second-generation dCas9-derived technologies, a side-by-side comparison and evaluation of 43 combinations of SunTag, SAM and scRNA-based transcriptional activation domains was conducted in tobacco [137]. Among all combinations assayed, the highest efficiency was achieved with a SAM system combining a dCas9-EDLL and several MS2 scRNA to recruit a tripartite activator VP64-p65-Rta. Finally, beside the omnipresent Class 2 dCas9, a new CRISPR-Cas system named CRISPR-associated complex for antiviral defense (Cascade) and based on a Class 1 CRISPR from *Streptococcus thermophilus* was developed. When transiently expressed in maize embryos, Cascade can effectively modulate gene expression by tethering the transcriptional activation domain from the *Arabidopsis* COLD BINDING FACTOR 1 (CBF1) [138].

### 6.3. dCas9-Based Epigenetic Editing

While sending transcription regulatory domains to specific target loci usually has profound indirect effect on chromatin structure and epigenetic marks, the dCas9 fusions can also be employed to directly modify specific epigenetic marks at the chromatin, by tethering enzymatic domains to target genes. Below, we provide a non-exhaustive list of successful dCas9-based epigenetic editing in animals, and present the few studies reported in plants thus far.

#### 6.3.1. Editing DNA Methylation

Among epigenetic processes, DNA-methylation/demethylation has early inspired dCas9-based systems to modulate transcription, particularly because of potential therapeutic applications [139]. For example, the full-length or catalytic domain of the *de novo* DNA methyltransferase DNMT3A in fusion with dCas9 or applied to the SunTag system can enhance precisely the CpG islands methylation at targeted genes, thus leading to reduced expression of those genes in various mammalian cell lines [140]. Multimerization of DNMT3a with the DNMT3L cofactor in fusion with the dCas9 can increase the edited methylation windows [141]. Interestingly, while a consistent off-target DNA methylation was detected with the dCas9-DNMT3A catalytic domain (DNMT3Acd) direct fusion, the SunTag-DNMT3Acd system exhibits much higher specificity and DNA methylation induction at the target sites [142]. Also, an engineered prokaryotic CpG-specific DNA methyltransferase MQ1 (i.e., derived from *Mollicutes spiroplasma*) fused to dCas9 can achieve efficient DNA methylation in mammalian cells more rapidly than the tools described above [143]. In *Arabidopsis*, a SunTag system with the methyltransferase domain of the tobacco DOMAINS REARRANGED METHYLTRANSFERASE (DRM) efficiently targeted DNA methylation at promoters of *FWA* and the floral development gene *SUPERMAN* (*SUP*) [136]. Conversely, the catalytic domain of TET1 DNA-demethylase (TET1cd) fused to dCas9, MS2, Casilio and SunTag can effectively induce demethylation as well as mRNA level increase at hypermethylated targeted genes in mammalian embryonic and cancer cell lines [141]. A SunTag-TET1cd system was also successfully implemented in *Arabidopsis* to target DNA demethylation and reactivate gene expression at the two methylated loci *FWA* and the *CACTA1* transposon with low levels of off-target effects [90]. Also, while DNA demethylation at *FWA* was complete and stably heritable in the absence of the transgene, re-silencing of *CACTA1* occurred when the corresponding transgene segregated away.

#### 6.3.2. Editing Histone Modifications

In addition to DNA methylation, a number of catalytic domains from histone-modifying proteins have been also used in several dCas9-related systems [141]. The histone demethylase LSD1 fused to dCas9 was able to repress two pluripotency maintenance genes in mouse embryonic stem cells by decreasing the levels of H3K4me2 and H3K27 acetylation near their respective enhancer regions [144]. To further expand the list of tools available for epigenetic silencing, catalytic domains from writers of H3K9me3 (G9A, SUV39H1) and full-length or catalytic domains from the writer of H3K27me3 (EZH2) directly fused to dCas9 were constructed to target three different promoters in two different cell types [145]. Surprisingly, while being sufficient for some level of repression with both cell type and/or target region variations, repression by chromatin writers was not always correlated with the deposition of expected or alternative repressive histone marks, suggesting non-catalytic mechanisms such as the simple steric interference. With a similar objective, a direct fusion between dCas9 and the full-length human histone deacetylase HDAC3 enabled repression of gene expression in mouse neuroblastoma cells in a target gene transcription level and acetylation status dependent manner [146]. More recently, the locus-specific manipulation of H3K27me3 for transcriptional repression in a living organism was established in Japanese killifish (*Oryzias latipes*) embryos using a fusion construct of the killifish H3K27 methyltransferase EZH2 and dCas9 [147]. For activating epigenetic modifications, the catalytic domain of the human acetyltransferase p300 directly fused to dCas9 was able to increase the level of H3K27ac at the enhancer and promoter of targeted genes, resulting in their transcriptional activation [79]. Using a gRNA library delivered via lentiviral infection, a dCas9-p300 activator was applied to the genome-wide functional identification of novel DNA regulatory regions [148]. Also with the purpose to upregulate target genes by inducing their intended histone mark around the transcription start site, catalytic domains from the human H3K4 methyltransferase PRDM9 and H3K79 methyltransferase DOT1L were shown to overcome epigenetic silencing at different transcriptionally repressed genes in various human cancer cell lines [133]. In particular, while H3K4me3 editing resulted in only transient re-expression of the DNA hyper-methylated target locus, co-targeting of PRDM9 and DOT1 effectors could maintain this de-repression [149]. In non-mammalian species, the direct fusion of dCas9 with the histone acetyltransferase domain CBP was reported to be a more potent activator than a SAM system targeting three different transcription activation domains in *Drosophila* cells. Interestingly, the opposite was observed at promoters or enhancers pre-marked with H3K27 acetylation [150].

In *Arabidopsis* transgenic plants, a scaffold MS2-based system recruiting three different histone-modifying domains was tested using the flowering time gene *FLOWERING LOCUS T* (*FT*) as a target [151]. Among the two H3K9 methyltransferase constructs, transformants expressing the catalytic SET domain of the human G9a displayed a wild-type phenotype, whereas transformants expressing the catalytic SET domain of KYP exhibited a late heritable flowering phenotype correlated with a decreased *FT* expression but no change in H3K9me2. *Vice-versa*, transformants expressing the catalytic domains of the human H3K27 acetyltransferase p300 were flowering earlier. However, while the level of H3K27 acetylation was increased, *FT* expression was unchanged and the early flowering phenotype was not inherited. Finally, as a promising strategy for improving stress tolerance in plants, a dCas9 epigenome editing using the *Arabidopsis* histone acetyltransferase 1 (AtHAT1) was reported to efficiently activate the endogenous promoter of *ABSCISIC ACID–RESPONSIVE ELEMENT BINDING PROTEIN1* (*AREB1*, also named *ABF2*), an important determinant in ABA signaling in stress-response [152]. By displaying a higher chlorophyll content and a faster stomatal aperture under water deficiency, transgenic *Arabidopsis* plants were more tolerant to drought stress.

#### 6.3.3. Pending Concerns with the dCas9 System

The newest dCas9 systems, while really promising for epigenetic editing, still present some drawbacks, which call for a thorough choice of the enzymatic module, cautious design of the gRNAs, and careful pick of the expression system (ubiquitous or specific promoter, constant or inducible expression or activity of the enzymatic module).

In order to epigenetically modify a sole dCas-targeted locus, the effector fused to dCas9 has to carry a well characterized enzymatic domain, avoiding any other unnecessary region of the chromatin modifier that would otherwise potentially target the fusion protein to undesired genomic locations. Thus, choosing enzymatic domains that have been structurally characterized may help to avoid undesired targets. Also, the precise spatiotemporal expression pattern of the target gene is a key parameter to take into consideration at the time of strategy design, in order to choose the most appropriate promoter for expression of the dCas9 system. Moreover, the likely transient character of gene expression changes brought by the epigenetic editing, needs to be taken into an account. Indeed, the dCas9-induced changes in gene expression are not always stable and heritable [90,151].

The epigenetic marks are part of complex and multivalent regulatory networks, making the interpretation of obtained results non trivial [151], but also raising the question of their maintenance. The use of an inducible epigenetic editing system may be instrumental in that sense, either driven by an inducible promoter (EtOH, estrogen), or containing an inducible enzymatic activity (e.g., fusion to the estrogen or glucocorticoid receptors). Such systems would allow repetitive inductions of the epigenetic editing module thanks to treatments with the corresponding compounds. The question of an induced spreading or locus-wide removal of an epigenetic mark also needs to be addressed, requiring the design of multiple gRNAs, in correspondence with the locus-based profile of the targeted mark. Another question is the efficiency of such tools when the target to be modified is under epigenetic repression and thus may have inaccessible chromatin in the endogenous genome. Indeed, it cannot be excluded that the poor accessibility of heterochromatin to the dCas9 fusion protein may hinder epigenome editing at these regions. Finally, the sole presence of dCas9 system components, with untargeted gRNA, may have undesired effects on gene expression, as reported in human cells [153]. Such side effects need to be taken into account in order to design all appropriate negative controls.

## 7. Concluding Remarks and Perspectives

The comprehensive studies of regulatory pathways require and stimulate the development of new, finer and flexible tools for targeted manipulation of gene expression and chromatin marks. With the development of dCas9-derived epigenetic edition tools, we have reached a breakthrough in approaches for depicting the function of epigenetic marks. Most often, these tools are first developed for application in mammalian cells, and are poorly or not yet tested in full animal organisms or tissues, nor in other model systems.

Refined causal studies for epigenetic mark effects on nuclear processes (such as transcriptional activation/repression, marks cross talks, chromatin remodeling/looping…) also require the building of inducible epigenetic editing tools. Such inducible tools have rarely been reported for their use in any model organism thus far, while others, because they are plant-based, present limitations for their application in plants. Among them is the CRISPR-Cas9-based Photoactivatable Transcription System [154]. It was designed to carry out a rapid and reversible gene activation in mammalian cells. The transcriptional activator, fused to the Cry2 domain is brought to the dCas9 bound region following the exposure to blue light. The CLOuD9 technology is another inducible system consisting of dCas9 proteins fused to a unique, reversible chemical induced proximity system utilizing the abscisic acid (ABA) plant phytohormone and components of the ABA signaling pathway [155,156]. It was used in order to manipulate the nuclear architecture through chromosomal looping, making use of two dCas9 variants each tethering one of the dimerizing ABA response proteins (ABI1 and PYL1). Finally, the FIRE-Cas9 is another inducible system, allowing the reversible recruitment of endogenous chromatin regulators upon rapamycine treatment. In this system, dCas9-MS2 recruits a MCP protein fused to rapamycine-dimerizable Fkbp/Frb proteins to enable chemical-induced proximity of a desired chromatin regulator [157]. This allowed to show that the recruitment of the SWI/SNF (BAF) chromatin remodeling complex led to activation of bivalent gene transcription in mouse embryonic stem cells. It also permitted to show that the recruitment of the Hp1/Suv39h1 heterochromatin complex resulted in the deposition of H3K9me3 and further gene silencing. Remarkably, these effects were reversible after removal of the chemical dimerizer [157]. Although not yet tested in plants, such a system opens a possibility for more flexible and conditional induction of the dCas9 activity and thus better resolution for studying the epigenetic state regulatory mechanisms.

In summary, the second generation dCas9 systems, together with the recent proof-of-concept studies on inducible/switchable systems, should allow identifying histone marks that can be manipulated at key genes to rewire transcriptional programs involved in specific aspects of plant development. Such strategies could be further translated to crops for adapting their development and growth to environmental constrains. The resulting discoveries will allow for the proposal of epigenetic rewiring strategies for important agricultural traits such as flowering time and flower or seed production.

## Figures and Tables

**Figure 1 ijms-22-00512-f001:**
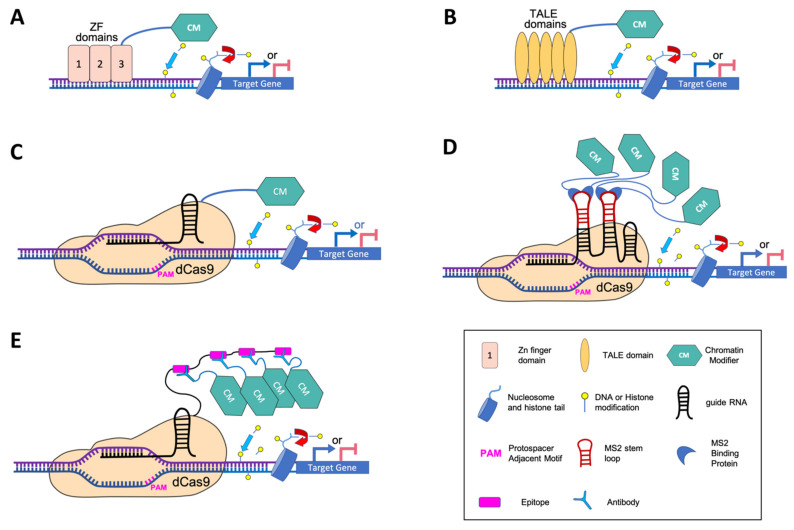
Overview of epigenetic engineering approaches to study gene function and chromatin modifications at specific loci. (**A**) ZF-based editing tool. In this approach, the fusion of several Zn finger domains forms a polydactyl system that targets a specific DNA sequence. Direct fusion to a chromatin modifier (CM) can then trigger chromatin modifications such as methylation/demethylation of either DNA or histones in nucleosomes, or acetylation/deacetylation of histones nearby the target site. (**B**) TALE-based editing tool. The TALE (Transcription Activator-Like Effectors) approach also uses customizable DBDs to target a specific DNA sequence as well as a direct fusion to a CM to induce chromatin modifications nearby the target site. (**C**) First generation of dCas9 tools to modify chromatin at specific loci. This approach is based on the property of a guide RNA (gRNA) to target a complementary DNA region of interest. The gRNA recruits the dead Cas9 (dCas9) protein which is directly fused to a CM. (**D**) Second generation of dCas9 tools with the MS2 strategy. The MS2 scaffold RNAs are recognized by MCP proteins fused to CMs, thus enhancing effector capacity. (**E**) Second generation of dCas9 tools with the SunTag strategy. The dCas9 is fused to a multicopy antigen which is recognized by an antibody fused to TMs or CMs, thus amplifying effector capacity.

**Table 1 ijms-22-00512-t001:** Chemicals tested in plants for chromatin modifications. This table focuses on the functional groups of chemical agents that were used to target the histone modifications in plants, by affecting the activity of histone deacetylases (HDACs), histone acetyltransferases (HATs), histone methyltransferases (HMTs), as well as disrupters of methyl supply. Table cells with green background correspond to chemicals which effect on histone marks was proven in plants; cells with white background correspond to chemicals which were tested in plants, but without specific proof of effect on marks. SAM: S-adenosylmethionine; SAH: S-adenosylhomocysteine. PABA: p-aminobenzoic acid (precursor of folates, which causes reduction in S-adenosylmethionine levels).

	Compound	Target *Origin*	Assays in Animals			Assays in Plants		
			Observed Effects	Techniques Used	Refs.	Observed Effects	Techniques Used	Refs.
**Inhibitors** **of** **HDACs**	**Trichostatin A (TSA)**	HDAC *Synthetic*	Differentiation of tumor cells in mammalian cell culture and rat mammary cancer Increase histone acetylation	Western blot, Immunoprecipitation, HDAC activity assay, Co-IP analysis	[21,22,23]	Affects global levels of histone acetylation and induces somatic embryogenesis in *Arabidopsis* Produces doubled haploid in wheat	RT-qPCR, microarrays, HDAC activity assay, fluorescent imaging quantification	[24,25,26,27,28]
	**Sirtinol**	Sirtuin-type HDAC *Synthetic*	Apoptotic and autophagic cell death in MCF-7 human breast cancer cells High inhibitor activity in leukemia cells	Western blot, flow cytometry	[29,30,31]	Impacts shoot and root meristems maintenance, affects body axis formation and vascularization in *Arabidopsis*	GUS activity measurement, RT-qPCR	[32,33,34,35]
	**Nicotinamide**	Sirtuin-type HDAC *Natural product of NADH_2_ oxidation*	Inhibitor of the SIRT1 in vitro. Affects H3K9 acetylation in rat brain cells	Western blot, RT-qPCR, MRI	[36,37]	Alters histone acetylation and induces VIN3 expression in *Arabidopsis*	RT-qPCR, ChIP-PCR	[38]
	**Ky-2, Ky-14Ky-9, Ky-72**	HDAC *Synthetic*	Affects inflammatory response in human macrophages. Enhances H3 acetylation in THP-1 cells	Western blot, RT-qPCR	[39,40]	Enhances high salinity stress tolerance in *Arabidopsis* and tobacco BY-2 cells through increase in H4 acetylation.Enhances H3 acetylation in *Arabidopsis*	RT-qPCR, Western blot, fluorescent imaging quantification	[26,41,42]
	**HC toxin**	HDAC *From Cochliobolus carbonum*	High efficiency in intrahepatic cholangiocarcinoma cells by inhibiting HDAC1 in a post-transcriptional manner	Flow cytometry analysis, Western blot, RT-qPCR, immunofluorescence	[43,44]	Leads to hyperacetylation of H4 and all isoforms of H3 histones in maize	HDAC activity assay, Western blot	[45]
	**Nitrtic oxide**	HDAC *From bacteria* (e.g., *Moraxella catarrhalis)*	Suppresses the serum-induced histone acetylation and enhanced histone deacetylase (HDAC) activity in human umbilical vein ECs (HUVECs)	Enzymatic activity assay, Western blot, immunofluorescence	[46,47]	Leads to hyperacetylation at specific genes in *Arabidopsis*	HDAC activity assay, Western blot, ChIPseq	[48]
	**Depudecin**	HDAC *From Alternaria brassicicola*	Morphological reversion of NIH 3T transformed fibroblasts	Trapoxin Binding Assay, histone acetylation assays	[49]	Has a minor role in virulence on Brassica oleracea but not in Arabidopsis	Discussed in the text but no supporting results	[50]
**Inhibitors** **of** **HATs**	**Compound C646, C107**	p300, acetyl-CoA competitor *Synthetic*	Inhibits histone H4 acetylation in animal cells	FRET, Western blot, RT-qPCR, radioactive assays of acetylation	[51]	Reduces the level of H3K9 acetylation in tobacco BY-2 cells and *Arabidopsis*	Western blot, fluorescent imaging quantification	[26,52]
	**Curcumin** (diféruloylméthane)	p300 *From Curcuma longa*	Inhibits histone acetylation in mammalian cells	Filter binding, fluorography, Hoechst staining Western blot	[53]	Affects H3 and H4 acetylation in maize and *Arabidopsis*	ChIP-PCR, RT-PCR, Western blot	[54,55]
	**MC1626, Anacardic acid** (quinolic analogue of anacardic acid)	p300 HAT *From cashew nut (Anacardium occidentale)*	Reversibly and noncompetitively inhibits HAT activity in *Plasmodium falciparum* Inhibits the H3 acetylation level in yeast	Western blot, ChIP-PCR, RT-qPCR, microarray	[56,57]	Inhibits UV-B induced deacetylation of H3K9 and H3K14 and the specific induction of UVR8-regulated genes in *Arabidopsis*	ChIP-PCR, RT-qPCR	[58]
	**MB-3, Gamma-butyrolactone**	mammalian GCN5 *Synthetic*	Inactivates the GCN5 in mammalian cell lines	RT-qPCR, Western blot, ChIP-PCR	[59]	Causes a decrease in H3K9 and H3K14 acetylation in *Arabidopsis*	ChIP-PCR, RT-PCR, Western blot, ChIPseq, RNAseq	[52,60]
**Inhibitors** **of** **HMTs**	**RDS 3434**, 1,5-bis (3-bromo-4-methoxyphenyl) penta-1,4-dien-3-one compound	SAM competitor for EZH2 binding *Synthetic*	EZH2 inhibitor in human leukemia cells	Western blot, qRT-PCR	[61]	Causes a decrease in H3K27me3 in a dose-dependent manner in *Arabidopsis*	RT-qPCR, Western blot	[62]
	**BIX-01294** (diazepin-quinazolin-amine derivative)	HMT *Synthetic*	Affects the H3K9me2 in mammalian cell lines	Immuno-cytochemical assay, RT-qPCR, Western blot, ChIP	[63]	Affects H3K9 methylation, promotes cell reprogramming, totipotency and embryogenesis in *Brassica napus* and *Hordeum vulgare*	Colorimetric histone methylation assay, immunofluorescence, RT-qPCR	[64]
**Disrupters of** **methyl supply**	**DHPA** (dihydroxypropyladenine)	SAH hydrolase inhibitor *Synthetic*	Inhibition of SAHH induced hypomethylation in the p66shc gene promoter in mice	Western blot	[65]	Induces decrease in DNA methylation and H3K9me2 and releases silenced transgenes in *Arabidopsis* and tobacco	ChIP-PCR, RT-qPCR	[66,67]
	**Sulfamethazine (SMZ)**	PABA competitive antagonist *Synthetic*				Reduces levels of DNA methylation and H3K9me2 in *Arabidopsis*	ChIP-PCR, RT-qPCR, bisulfite sequencing	[68]

## Data Availability

Not applicable.

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
