# Peer review of "Chromatin Manipulation and Editing: Challenges, New Technologies and Their Use in Plants"

_ijms, 2021, doi:10.3390/ijms22020512_

Round 1

Reviewer 1 Report

The review manuscript by Fal, Tomkova and colleagues entitled “Chromatin manipulation and editing in plants: Challenges, approaches and new technologies” present available tools for epigenome editing in plants. Considering the recent successful reports of epigenome editing using the modified CRISPR/Cas9 strategy and the potential of such approaches to resolve long-lasting questions in chromatin biology and to produce transgene-free improved crops, this review is of high interest and particularly timely. The manuscript is well organized, clear and comprehensive.

I have a few comments and a few suggestions that could improve the manuscript, detailed below.

Line 39: NRLs depend on cell type and species as stated but also, inside a given cell, it depends greatly on the underlying genomic regions and sequences (longer NRL in heterochromatin for example).

Line 86: I agree on “correlative” but I find “only limited” too strong considering all the important discoveries made by mutating chromatin readers.

Line 149: The advantage of chromatibody over tagged histones should be specified.

Line 228-229: I would rephrase this sentence. First, the CRISPR-Cas9 system can be powerful enough to get multiple mutations at once (several examples of sextuple mutants in Arabidopsis have been published). Second, some histone variants have few copies of in the genome such as H1, H2A.W or H2A.Z. Third, a limitation would rather be the potential lethality of such mutants, as shown for H3.3 (See https://doi.org/10.1186/s13059-017-1221-3).

Line 308: The first article cited for ZNF targeting of a chromatin modifier is the article about TET1 by Gallego-Bartolomé et al. However, an earlier successful report by the Jacobsen lab was the tethering of SUVH2 (https://doi.org/10.1038/nature12931). This was to my knowledge the first example of epigenome editing in plants and should be referenced in this review.

General comments on Part 4:

The organization of this part could be improved.

It is entitled “Direct gene manipulation to modulate chromatin genome-wide”, the first subsection is about mutating histone modifiable residues and the second about mutating chromatin factors by Cas9.

This organization elude all the discoveries made through classical mutation strategies (such as all the T-DNA insertion mutants of chromatin genes, even if as stated in the introduction, they reached their limit), without stating exactly what would be the benefits now of using the CRISPR-Cas9 tool to mutate chromatin modifiers in plants. If using T-DNA mutants for chromatin modifiers has reached its limit, using mutants for those genes produced by Cas9 should have too. This should be clarified in the text.

Accordingly, this Part 4 would be better if divided in methodological sections rather than aims, for clarity and also for consistency with the rest of the manuscript. For example, but it is only a suggestion:

  • Limitations of T-DNA insertion lines (a few successful specific cases could be discussed, such as H2B mono-ubiquitination, the knock-down of H3.3 or H2A.W and Z mutants before explaining limitations)
  • Expressing a mutated transgenic version of chromatin genes (with the specific focus on histone modifiable residues)
  • New perspectives brought by the Cas9 editing tool

Line 258: “at will,” should be removed

Line 339: I did not understand “constitutive and drought-induced”.

Line 397: Plants instead of plans.

Page 11-12 and half of 13: This is to present how the second generation of Cas9 was improved and it is interesting. But it is all about transcription factors and not about epigenetics. In consequence I would suggest shortening all this part. The same holds true about the parts on ZNF and TALEs dedicated to artificial TFs.

Line 494: enhance instead of enhanced

Line 589-597: Even if obvious, maybe it should be stated that such inducible systems cannot be used in plants as they are plant-based.

The text of part 5 and 6 could be shortened and made more comprehensive by adding a second table to the review with the epigenetic marks that have been successfully engineered by ZF, TALE and dCas9 in the described systems and in plants.

Figure 1: The review being specifically about epigenome editing I advise removing panel B and C that are unrelated. Make panel E and F only about chromatin modifiers (CM) and remove Transcriptional Regulators (TR). I would split panel A in 2 for ZNF and TALE and make them also uniquely about chromatin modifiers.

I feel that the drawbacks and difficulties of the dCas9 system is not enough discussed. For example, the poor accessibility of heterochromatin to the Cas9 protein that may hinder epigenome editing at these regions. Or, is the dCas9-chromatin modifier fusion protein targeting only the desired locations and really not able to bind to all the genomic locations the chromatin modifier alone would normally be targeted to? Finally, how many guides are needed for efficient targeting, the things to be considered when choosing the dCas9 promoter etc.

Finally, a sentence on the potential of epigenome editing for non-GMO crop improvement could be added to the conclusion. Some attempts of a direct delivery of Cas9/gRNA ribonucleoproteins have been reported, thereby producing transgene-free modified crops.

Reviewer 2 Report

This manuscript provides an overview of recent tools engineered for chromatin
manipulation and editing. In addition to describe the different molecular tools and
strategies, the authors compare the various technologies and highlight their associated
advantages and limits in both animals and plants. The manuscript is very well written. I
found this review really clear, informative, and pleasant to read. Please find below my
comments and suggestions.
- Title: As the authors describe the chromatin manipulation and editing in plants
and animals throughout the manuscript, I would suggest to include “animals” in
the title. The title could be “Chromatin manipulation and editing in plants and
animals: Challenges, approaches and new technologies”.
- Figure 1: “PAM” should be explained in the legend.
- Figure 1A: The meaning and application of “TALEN” should be explained in the
text.
- Line 556: “faster stomatal aperture” should be replaced by “faster stomatal
closure” I think. Please see the results of the reference [195]: “1h after drought
stress, the stomatal aperture was 1.6-fold lower in dCas9HAT-sgA2 plants than in
control plants (Fig. 4D). After 2 h, the stomatal aperture was still significantly
lower in dCas9HAT-sgA2 plants, even though control plants also partially closed
their stomata between 1h and 2h of stress. However, our data revealed that
stomatal closure was triggered more rapidly in dCas9HAT-sgA plants. After 20
days of stress, the stomatal aperture of the dCas9HAT-sgA plants was comparable
to that of the control plants, suggesting that dCas9HAT-sgA2 does not regulate
stomatal aperture differently during long-term drought stress. Our results
indicate that the expression of dCas9HAT-sgA2 leads to faster stomatal closure
after severe drought stress and corroborate a previous study stating that AREB1
might be partially associated with stomatal closure”.
- Table 1: The name of species should be in italic.
- Line 188: “A global reduction of H3K36 methylation” was already mentioned line
186. Please revise the writing as needed.
- Lines 308-309: “Firstly, a fusion of the TET1 catalytic domain with an artificial ZF
designed to target either the promoter of the FLOWERING WAGENINGEN (FWA)
gene or the CACTA1 transposon.” The verb needs to be corrected.
- Line 397: “Single gRNA-targeted transcriptional activation or repression was also
achieved in plans”. I guess “plans” should be replaced by “plants”.
